# Bio-Inspired Rhamnolipids, Cyclic Lipopeptides and a Chito-Oligosaccharide Confer Protection against Wheat Powdery Mildew and Inhibit Conidia Germination

**DOI:** 10.3390/molecules27196672

**Published:** 2022-10-07

**Authors:** Nour El Houda Raouani, Elodie Claverie, Béatrice Randoux, Ludovic Chaveriat, Yazen Yaseen, Bopha Yada, Patrick Martin, Juan Carlos Cabrera, Philippe Jacques, Philippe Reignault, Maryline Magnin-Robert, Anissa Lounès-Hadj Sahraoui

**Affiliations:** 1Unité de Chimie Environnementale et Interactions sur le Vivant (EA 4492), Université Littoral Côte d’Opale, CEDEX CS 80699, 62228 Calais, France; 2Materia Nova ASBL, Avenue du Champ de Mars 6, 7000 Mons, Belgium; 3ULR 7519—Unité Transformations & Agroressources, Université d’Artois, UnilaSalle, CEDEX CS 20819, 62408 Béthune, France; 4Lipofabrik, Parc d’Activités du Mélantois, 917 Rue des Saules, 59810 Lesquin, France; 5Fyteko, Allée de la Recherche 4, 1070 Brussels, Belgium; 6JUNIA, Joint Research Unit UMRt 1158-INRAE, BioEcoAgro, Équipe Métabolites Spécialisés d’Origine Végétale, University Lille, INRAE, University Liège, UPJV, University Artois, ULCO, 48, Boulevard Vauban, CEDEX BP 41290, 59014 Lille, France; 7Joint Research Unit 1158 BioEcoAgro, Équipe Métabolites Spécialisés d’Origine Végétale, Microbial Processes and Interactions, TERRA Research Centre, Gembloux Agro-Bio Tech, Université de Liège, 5030 Gembloux, Belgium

**Keywords:** wheat, powdery mildew, fungal disease, cyclic lipopeptides, chitosan, rhamnolipids, biocontrol, *Blumeria graminis* f.sp. *tritici*

## Abstract

Plant protection is mainly based on the application of synthetic pesticides to limit yield losses resulting from diseases. However, the use of more eco-friendly strategies for sustainable plant protection has become a necessity that could contribute to controlling pathogens through a direct antimicrobial effect and/or an induction of plant resistance. Three different families of natural or bioinspired compounds originated from bacterial or fungal strains have been evaluated to protect wheat against powdery mildew, caused by the biotrophic *Blumeria graminis* f.sp. *tritici* (*Bgt*). Thus, three bio-inspired mono-rhamnolipids (smRLs), three cyclic lipopeptides (CLPs, mycosubtilin (M), fengycin (F), surfactin (S)) applied individually and in mixtures (M + F and M + F + S), as well as a chitosan oligosaccharide (COS) BioA187 were tested against *Bgt*, in planta and in vitro. Only the three smRLs (Rh-Eth-C12, Rh-Est-C12 and Rh-Succ-C12), the two CLP mixtures and the BioA187 led to a partial protection of wheat against *Bgt*. The higher inhibitor effects on the germination of *Bgt* spores in vitro were observed from smRLs Rh-Eth-C12 and Rh-Succ-C12, mycosubtilin and the two CLP mixtures. Taking together, these results revealed that such molecules could constitute promising tools for a more eco-friendly agriculture.

## 1. Introduction

Wheat (*Triticum aestivum* L.) is one of the most produced cereal crops in the world and is responsible for feeding nearly 35% of the global population [1]. In regard to the world population growth projection to reach 9 billion people in 2050, mentioned by the Food and Agriculture Organization of the United Nations [2], wheat production should increase by around 60% to satisfy the future demand [3]. Wheat is susceptible to a great number of fungal pathogens, including *Blumeria graminis* f.sp. *tritici* (*Bgt*), responsible for powdery mildew (PM). This foliar disease can cause yield losses up to 34% [4] and is usually the first disease to occur during wheat cultivation [5]. *Bgt* is an obligate parasite that requires a living host to grow and reproduce. PM impact is becoming more severe, linked to the use of dense plant canopy of modern cultivars and also the use of high nitrogen fertilization, two factors known to favor PM development [6,7]. Development of resistant varieties and chemical control by application of conventional fungicides are the control strategies most commonly used for PM management. The PM-resistance (Pm) genes are the prerequisite for developing resistance wheat cultivars. To date, 91 Pm genes have been identified on 54 loci of wheat chromosomes [8]. However, the rapid appearance of new virulent races of *Bgt* often causes the circumvention of resistance of wheat varieties in a relatively short period of time [3]. The consistency and the severity of the damages caused by *Bgt* render necessary a systematic and extensive used of conventional pesticides. Again, some *Bgt* populations can become resistant to various pesticides such as benzimidazoles [9], sterol 14α-demethylation inhibitors [10] or strobilurins [11]. Moreover, the suspicion of the serious and long-term risks of chemical pesticides on human health [12] and environment [13,14] leads to the development of new strategies compatible with a sustainable agriculture.

When a plant is attacked by a pathogen, the activation of defense responses implies the essential step of the microorganism perception by highly conserved molecular patterns called PAMPs (Pathogen Associated Molecular Patterns) or MAMPs (Microbe Associated Molecular Patterns), which are produced by microorganisms or released from their cell wall by plant hydrolytic enzymes during interaction with the host [15,16,17]. Such PAMPS, considered as general elicitors, are represented by structurally distinct molecules, including well known components such as flagellin peptides, chitin, or β-glucans [18] and also bacterial biosurfactants, rhamnolipids and cyclic lipopeptides [19]. These molecules could be considered as interesting tools for crop protection. 

Much attention is reported on carbohydrate-type PAMPs, such as chitin, a linear polysaccharide polymer composed of D-glucosamine (GlcN) and N-acetyl-D-glucosamine (GlcNAc) linked by β-(1,4) covalent bonds [20]. Chitin is synthesized by a large number of living organisms, in particular by the filamentous fungi where chitin represents 10–20% of biomass but also by insects and crustaceans [21]. Chitosans, obtained by partial deacetylation of chitin, are well known to induce plant resistance to various pathogenic microorganisms (bacteria, fungi, oomycetes) on various crops, monocotyledons or dicotyledons, herbaceous or lignified plants [22,23]. The perception of chitosan oligosaccharides (COS) by plants is mediated by a molecular pattern recognition receptor (PRR). In Arabidopsis, lysine-motif receptor kinases (LysM-RLK) such as CERK1, LYK4, and LYK5, act as chitin PRRs to regulate the chitin signaling pathway [24,25,26]. A recent study demonstrated that dimerization of CERK1 is required to activate immune signaling [27,28]. After perception, a complex of signaling reactions involved in chitosan-mediated signal transduction have been identified, such as the rapid accumulation of hydrogen peroxide [29], a production of nitric oxide [30], and a transient increase in cytosolic calcium concentration [31], which lead to the modulation of phytohormone levels in plant tissues [32,33] and the induction of plant defense reactions [23,30,34]. Moreover, the antifungal activity of chitosan compounds has been reported against various phytopathogenic fungi such as *Botrytis cinerea*, *Rhizopus stolonifer*, *Colletotrichum* sp., *Sclerotinia sclerotiorum*, *Rhizoctonia solani*, *Fusarium oxysporum* and *Verticillium dahliae* [35,36,37,38,39,40]. This biological activity was characterized by an inhibition of mycelial growth, sporulation or spore germination [37,39,41,42].

An interesting dual activity was also observed for non-ribosomal cyclic lipopeptides (CLPs); these ones are composed of a lipid tail linked to a cyclic oligopeptide and are widely produced by various bacterial species often referred to as plant-beneficial bacteria [19,43,44]. According to their amino acid sequence, these cyclic lipopeptides are divided into three families: iturins (mycosubtilin (M), iturin A, and bacillomycin), surfactins (S), and fengycins (F) [45,46]. Members of iturin and fengycin families are mainly characterized for their strong antifungal activities on phytopathogenic agents [47], while molecules of the surfactin family are mainly known to induce systemic resistance (ISR) in plants [43,48]. Concerning the effects of CLPs on wheat pathogenic fungi, Mejri et al. [49] suggested a strong antifungal activity of mycosubtilin against *Zymoseptoria tritici*, the causal agent of septoria tritici blotch, characterized by the inhibition of fungal growth in vitro and on the wheat leaf surface. An important antifungal activity of the iturin A has been demonstrated against *Fusarium graminearum*, responsible for fusarium head blight of wheat, resulting in an inhibition of spore germination, a strong inhibition of hyphal growth and a loss of plasma membrane integrity [50]. 

Other bacteria produce rhamnolipids (RLs), which are glycolipidic biosurfactants and consist of rhamnose units, glycosidically linked to long-chain hydroxyl fatty acids consisting of 8 to 16 carbon atoms [51]. RLs are naturally produced by various bacteria including the genus *Pseudomonas* sp., *Burkholderia* sp., *Acinetobacter* sp. or *Pantoea* sp. [52] and could be used in agriculture for crop protection [19,53]. RLs are naturally produced as a mixture of mono-rhamnolipids and di-rhamnolipids for which the mass ratio of mono-rhamnolipids to di-rhamnolipids can vary with the fermentation time [45,54]. The RLs exhibit antimicrobial properties against a large panel of fungal phytopathogens [19] and are also known for the stimulation of plant immunity [19,55]. A crude extract of rhamnolipid-containing cell-free culture broth from *P. aeruginosa* ZJU211 (mixtures of mono and di-rhamnolipids) showed a strong antifungal activity against *F. graminearum* with a substantial decrease of fungal growth and biomass accumulation [54]. 

The objective of the present study was thus to assess on the wheat-*Bgt* pathosystem the direct antifungal activity as well as the protection efficacy of various compounds isolated from microorganisms or bio-inspired. One COS, BioA187, was obtained by enzymatic hydrolysis of a commercial fungal chitosan and specifically prepared for this study; three CLPs (mycosubtilin, fengycin and surfactin) and two mixtures of them (M + S, M + F + S) were produced by various *B. subtilis* strains at the bioreactor level, and three smRLs (Rh-Eth-C12, Rh-Est-C12 and Rh-Succ-C12) were synthetized by green chemistry. 

## 2. Results

### 2.1. Characterization of the COS BioA187

The chemical structure of BioA187 was confirmed by FTIR, and a strong broad absorption peak at 3350 cm^−1^ was an overlap of N–H and O–H stretching vibration absorption peaks, which were sensitive to intermolecular hydrogen bonds (Figure 1). Peak at 2884 cm^−1^ was the absorption of C–H stretching vibration of methyl or methine. The FTIR spectrum of COS displayed peaks at 1634, 1529, and 1378 cm^−1^, corresponding to characteristic absorption peaks of amide I, amide II, and amide III bands. Furthermore, the absorption peak at 1073 cm^−1^ was the stretching vibration of C–O in the C-6 position. As observed on thermogram in Figure 2 of TGA analysis, in the interval between 30 °C and 100 °C (63 °C), a weight loss is attributed to water evaporation. Then the weight of partially deacetylated chitosans remained stable up to 250 °C followed by a very strong loss of weight up to a maximum loss at 334 °C. The TGA thermogram showed a 54% weight loss corresponding to the pyrolytic decomposition of the molecule at 283.6°C, the maximum degradation temperature of BioA187 (red line in Figure 2). According to Corazzari et al. [56], this corresponds to a main process involving the release of H_2_O, NH_3_, CO, CO_2_ and CH_3_COOH in the temperature range 250–450 °C, causing a weight loss of 54% and corresponding to the pyrolytic degradation of the chitosan characteristic of the chitosan structure [57,58]. The DD of COS BioA187 was evaluated at 63.3%, which represents a relatively high DD.

The MALDI-TOF-MS of COS BioA187 is shown in Figure 3 and more information about the assigned structure of each signal is given in Table 1. It is very clear from Figure 3 that the strongest *m/z* signals originate from fully deacetylated chitosan oligosaccharides of degree of polymerizations (DPs) 3–9 but *m/z* signals corresponding to oligomers of higher DPs are also detected.

### 2.2. The COS BioA187, the Three Bio-Inspired smRLs and Mixtures of CLPs Conferred Partial Protections of Wheat against B. graminis f.sp. tritici 

Three smRLs (Rh-Eth-C12, Rh-Est-C12, Rh-Succ-C12), the COS BioA187 as well as bacterial CLPs treatments (M, S, F and two mixtures M + S, M + S + F) were tested in controlled conditions for their protection efficacy against *Bgt* on one susceptible cultivar of wheat (Pakito). Disease symptoms were observed on control plants (0.1% DMSO-treated) and evaluated with an average of 185 colonies on the third leaf (Figure 4). Spraying of the three smRLs, Rh-Eth-C12, Rh-Est-C12 and Rh-Succ-C12, at 100 µM conferred a significant protection to wheat against *Bgt* by about 55, 60 and 65%, respectively, compared to the control. BioA187 also led to a significant wheat protection against *Bgt* by about 47%. Concerning the tested CLPs, only the treatments combining mycosubtilin and surfactin (M + S) and mycosubtilin, surfactin and fengycin (M + S + F) significantly protected the plants, by 49 and 54%, respectively (Figure 4).

### 2.3. Antigerminative Effect of COS BioA187, the Three Bio-Inspired smRLs and CLPs on B. graminis f.sp. tritici Spores

The impact of molecules and mixtures on spore germination was observed in vitro. The rate of non-germinated spores and spores with aborted tubes (C1 class) reached 57, 79 or 62% for each repetition of the control condition (0.1% DMSO in agar, Figure 5a, Figure 5b and Figure 5c, respectively).

Concerning the effect of smRLs on *Bgt* spore germination, Rh-Eth C12 totally inhibited spore germination at all tested concentrations (Figure 5a). At the highest concentration of Rh-Est C12 (300 µM), all spores either failed to germinate or showed aborted tubes. Rh-Est C12 at 100 and 200 µM appeared to increase the rate of non-germinated and germinated spores with aborted germinative tubes (Class 1) by 56 and 63% in comparison to the control, when no significant effect on spore germination was observed at 50 µM (Figure 5a). The inhibitory effect of Rh-Succ C12 on spore germination was observed from 50 µM, with a significant increase of the rate of non-germinated spores and spores with aborted tubes by 58, 61, 68 and 72% for 50, 100, 200 and 300 µM, respectively (Figure 5a).

No effect on spore germination was observed in response to BioA187 treatment at 50 µM (Figure 5b). For higher concentrations (100 to 300 µM), a significant effect was observed, with an increase of 20% of C1 *Bgt* spores comparing to control (Figure 5b). Finally, regarding the effect of lipopeptides in individual treatment, the mycosubtilin showed a significant inhibitory effect at all the tested concentrations, with a total inhibition of *Bgt* spore germination from 100 µM (Figure 5c) when at 50 µM a significant increase of the non-germinated spores and spores with aborted tubes was evaluated at 49% (Figure 5c). No effect on spore germination was observed in response to fengycin treatment at 50 µM. Fengycin showed an increase of the C1 *Bgt* spores at 100 µM comparing to control by 44%, when a total inhibition of spore germination is observed from 200 µM. In contrast, surfactin did not show such a significant inhibitory effect at 50 or 100 µM. A significant effect was observed from 200 µm, with C1 *Bgt* spores evaluated at 80 and 83.3% for 200 and 300 µM in comparison to the control, respectively. Finally, the effect of both mixtures of lipopeptides (M + S or M + F + S) resulted in a total inhibition of spore germination from 100 µM, similar to that obtained in the presence of mycosubtilin alone (Figure 5c). An important increase of the C1 *Bgt* spores was also observed at 50 µM for both mixtures, by 45 and 58% compared to the control for M + S or M + F + S, respectively.

## 3. Discussion

Fungal disease control strategies rely mainly on the use of chemical fungicides, resistant cultivars and to a lesser extent on adequate cultural practices. However, the current concern on the extensive use of conventional pesticides and the frequent adaptation of fungal populations make urgent the development of new alternative control strategies compatible with a more sustainable agriculture. In recent decades, eco-friendly and biodegradable compounds isolated from microorganisms have been considered for biocontrol strategies, since they can act through antimicrobial activity or by stimulating the plant immune system to sensitize the plant to a subsequent infection [19,23]. In the current study, we investigated the use of several compounds extracted from bacterial or fungal microorganisms, as well as synthetic bio-inspired RLs, on wheat against *Bgt*. Our results revealed that all treatments tested, i.e., the bio-inspired smRLs, the fungal COS BioA187 and bacterial LPs alone or in mixture resulted in partial or total inhibition of *Bgt* spore germination in vitro, depending on the concentrations tested. Interestingly, five of them also reduced the expression of powdery mildew symptoms on young wheat seedlings, i.e., smRLs Rh-Eth-C12, Rh-Est-C12, the two LP mixtures and COS BioA187, with very close protection rates.

RLs are naturally produced by various bacterial species including some *Pseudomonas* and *Burkholderia* species [52]. They have already been reported to protect plants against fungal pathogens. Platel et al. [59] tested various bioinspired smRLs with a 4, 8, 12, 14 and 16 carbon fatty acid tail against *Z. tritici* during in vitro and in planta assays and revealed that RLs with a 12-carbon fatty acid tail were the most effective. In our study, a foliar spraying of Rh-Eth-C12, Rh-Est-C12 and Rh-Succ-C12 also led to a significant reduction in powdery mildew symptoms on wheat, which was over to 50% when used at 100 µM. Taken together, these results highlight the importance of the C12 chain in the protective activity of RLs on wheat against both powdery mildew and septoria tritici blotch diseases. However, the function (ether, ester or succinate) linking the hydrophobic chain does not influence the protection efficiency obtained in our pathosystem. Indeed, very close protective rates, 55, 60 and 65%, were obtained against *Bgt* for Rh-Eth-C12, Rh-Est-C12 and Rh-Succ-C12, respectively. On the contrary, only foliar applications of Rh-Eth-C12 and Rh-Est-C12 at the concentration of 1.5 mM led to a significant protection of the wheat against *Z. tritici* [59]. The protection rates obtained were significantly different, 15.5% and 35.6% for Rh-Eth-C12 and Rh-Est-C12, respectively. Moreover, these authors clearly demonstrated that wheat protection rates against *Z. tritici* in response to foliar application of Rh-Est-C12 were concentration-dependent, with protection rates evaluated at 19.9, 53.3, and 78.9% with application of 800, 1600, and 3200 µM of Rh-Est-C12, respectively [59]. In tomato, the application of Rh-Eth-C12 and Rh-Est-C12 at 300 µM resulted in a significant reduction of leaf necrosis induced by *Botrytis cinerea* [60]. In the present study, a direct antifungal activity on *Bgt* spore germination was observed for the three smRLs tested, with a higher anti-germinative activity of Rh-Eth-C12, which is observed from the concentration of 50 µM. Nevertheless, only Rh-Eth-C12 and Rh-Est-C12 appeared to significantly reduce *Z. tritici* growth, in vitro, while no direct activity was detected for Rh-Succ-C12 [59]. Another recent study demonstrated an inhibition effect on *B. cinerea*, through a reduction of conidial germination and alteration of mycelium growth in vitro conditions in response to various smRLS, such as Rh-Eth-C12 and Rh-Est-C12 [60]. Moreover, this work described a curative effect of Rh-Eth-C12 [60]. Taken together, these results confirmed the antifungal activity of these two smRLs on various phytopathogenic fungi. Little data are available to date to understand how these compounds lead to the inhibition of spore germination or fungal growth, but with regard to their amphiphilic properties, they could interfere with the fungal cell membrane as suggested for natural RLs [61]. Moreover, various studies suggested that smRLs, such as Rh-Eth-C12 and Rh-Est-C12, might increase resistance of plants against fungal pathogen by dual activities combining antifungal properties and induction of plant defense mechanisms [59,60,61], as has previously described for natural rhamnolipids [19].

Cyclic lipopeptides are other bacterial-produced compounds that could be used against fungal diseases since they were demonstrated to be less ecotoxic in comparison to chemical fungicides [62]. Our results also revealed that the use of CLPs may also be developed against powdery mildew as a new alternative control strategy since the majority of CLPs tested alone or in mixtures (excepted for surfactin alone) displayed high and very close antifungal effect on *Bgt* spore germination in vitro, while only the two mixtures containing mycosubtilin+surfactin or mycosubtilin+surfactin+fengycin at 100 µM led to a significant reduction of *Bgt* symptoms on wheat plantlets.

In regard to these findings, we first confirmed the antimicrobial activity of members of iturin and fengycin families on fungal phytopathogens [19,47]. Such an antifungal activity appear to be related to the ability of the CLPs to form pores in the fungal cell membrane [40]. Mejri et al. [49] demonstrated a high ability of mycosubtilin to reduce spore germination of *Z. tritici* both in vitro and in planta compared to fengycin or surfactin treatments. Cao et al. [63] suggested that the relative contribution of different LPs to antimicrobial activity may be dependent on the species of plant pathogen encountered. These authors suggested that iturins and fengycins are functionally redundant in the antagonism against the bacterium *Ralstonia solanacearum*, when only iturin family metabolites appear to display antagonistic activity against the fungus *F. oxysporum*.

Moreover, the protective activity observed only in response to mixture applications suggests that they could increase resistance of wheats against *Bgt* by combining a direct antimicrobial effect of mycosubtilin or/and fengycin and the induction of plant defense mechanisms by surfactin. Considering CLPs, surfactin is mainly known to activate ISR via its perception in the lipid bilayer membrane of plants cells, which acts as a sensor [64]. Various compounds of the surfactin family are known to induce plant defense [65] leading to the protection of tomato against *B. cinerea* [66], grapevine against *Plasmopara viticola* [67], peanut against *Sclerotium rolfsii* [68] or also wheat against *Z. tritici* [69]. Moreover, few studies also reported that fengycin and also mycosubtilin stimulate the induction of plant defense reactions in the plant [70,71]. Another hypothesis could also explain the increased performance of mycosubtilin/fengycin driven by surfactin. Indeed, surfactin has surface-active properties that could increase the penetration and foliar retention of mycosubtilin or the mycosubtilin/fengycin mixture, optimizing their antifungal effects [64,65]. Likewise, Deravel et al. [62] reported a similar protective activity of mycosubtilin/surfactin mixture on lettuce against the biotrophic fungus *Bremia lactucae* compared to a mycosubtilin treatment alone, but with a mycosubtilin concentration twice smaller in the mix than in the mycosubtilin treatment alone.

Plant protection against fungal diseases using chitosan-based solutions has been extensively reported in various crops, such as grapevine [72], tomato [73], cucumber [36], barley [74] and also wheat [75]. Chitosans appeared as promising alternative control products since these compounds exhibit complete biodegradability and low toxicity [76,77,78,79]. BioA187 studied here has been obtained by enzymatic hydrolysis of fungal chitosan, which is an effective and safety COS preparation method [80]. The protection conferred in response to a chitosan treatment would result not only from its direct inhibitory activity but also from the stimulation of the plant defenses [72,81]. In our results, a partial anti-germinative activity on *Bgt* spores was observed with BioA187 from 100 µM. This direct activity of the COS BioA187 confirms studies showing an anti-germinative action on different phytopathogenic fungi, such as *B. cinerea*, *Alternaria radicina*, *Penicillium expansum* [36,82,83]. Nevertheless, an absence of direct inhibitory effect of some chitosan-based products against other fungal species was also observed and could be explained by the fact that the direct toxicity of chitosans remains dependent on their molecular weight, degree of acetylation, solvent, pH and viscosity [22,84,85]. Moreover, Park et al. [86] suggested that their inhibitory activity was dependent on the fungal species tested. The main chemical properties conditioning chitosan biological activities are their degree of deacetylation (DD) and molecular weight, which for chitooligomers is also expressed as degree of polymerization [87,88]. The DD is defined as the molar fraction of GlcN in the copolymers (chitosan) composed of GlcNAc and GlcN. Younes and Rinaudo [89] reported that chitosan with a higher DD showed stronger biological effects as well as an increased water solubility, due to a higher concentration of amino groups in the molecule, and that the protonation of the -NH_2_ functional group is vital for manifesting chitosan’s biological effects and water solubility [90]. BioA187 presents a relatively high DD (63.3%), which could explain its anti-germinative action on *Bgt* spores. The protective activity against *Bgt* observed in planta in response to the treatment by BioA187 could be associated to the anti-germinative activity of this chitosan-derived product. However, the contribution of a plant defense stimulation cannot be ruled out. Chitosans are indeed known to induce plant resistance to various pathogenic microorganisms (bacteria, fungi, oomycetes) on various crops [21,22,23,91]. In the case of wheat, a pretreatment of seedlings with chitosans control the *Fusarium graminearum* (agent of fusarium head blight) infection by stimulating the accumulation of phenolics and lignin [75].

Our results indicated that the tested compounds, extracted or inspired from bacterial or fungal micro-organisms, are promising products to manage fungal diseases on wheat and are compatible with a more sustainable agriculture. Thus, the three smRLs, the two CLP mixtures (M + S and M + F + S) and chitosan-derived BioA187 allow us to reach a partial protection of wheat higher than 50% against *Bgt* under controlled conditions. These promising molecules may confer protection to wheat through a direct inhibition of spore germination and plant defense elicitation. Regarding the antimicrobial activity of the compounds and the partial protection obtained, a spraying application of treatments combining these different molecules could be an interesting solution to optimize the protection rate of wheat against powdery mildew. Indeed, Marangon et al. [92] have reported an increase of antimicrobial activity by combining chitosan and rhamnolipid. Furthermore Varnier et al. [93] observed a potentiation of plant defenses induction by the chitosan with a pre-treatment of grapevine cell suspensions with RLs. The stimulating effects on plant cells clearly depends on the perception of these compounds. RLs and CLPs may be perceived through an interaction with the plasma membrane of this compound, as an initial stimulus triggering the downstream defense reactions [19]. The perception of chitosan oligomers by plants is rather known to be mediated by specific PRRs known as lysine-motif receptor kinases [24,25,26]. Thus, it would be also very interesting to investigate the capacity of wheat to perceive the compounds, activate its defense reactions and induce resistance in response to these treatments, applied individually and especially in mixtures.

## 4. Materials and Methods

### 4.1. Bio-Inpired Synthetic Mono-Rhamnolipids (smRLs)

A total of 3 smRLs with a 12-carbon fatty acid tail, and varying in their linkers, i.e., Rh-Eth-C12 (Dodecyl α/β-L-rhamnopyranoside), Rh-Est-C12 (Dodecanoyl α/β-L-rhamnopyranoside), and Rh-Succ-C12 (Dodecenylsuccinate α/β-L-rhamnopyranoside), were synthetized using green chemistry (Figure 1). The general procedures for rhamnose ether, ester and mono-rhamnosyl (alkenyl) succinate molecule synthesis have been reported in Robineau et al. [60]. The conversions and purities of the synthesized compounds were determined by NMR spectroscopy. NMR spectra were recorded on a Bruker DRX400 spectrometer (Bruker Biospin, France) operating at 400 MHz for 1H nuclei and 100 MHz for 13C nuclei. CDCl3 (99.50% isotopic purity) were purchased from Euriso-Top (Saarbrücken, Germany). The progress of the reactions was checked by thin-layer chromatography (TLC) on Merck silica gel 60 glass plates. Detection was carried out by spraying the chromatograms with 10% sulfuric acid in ethanol and heating them to 100 °C. Flash column chromatography was performed with silica gel (40–100 lm, Merck, Molsheim, France): all chemicals were of reagent grade and used without further purification.

**Scheme 1 molecules-27-06672-sch001:**
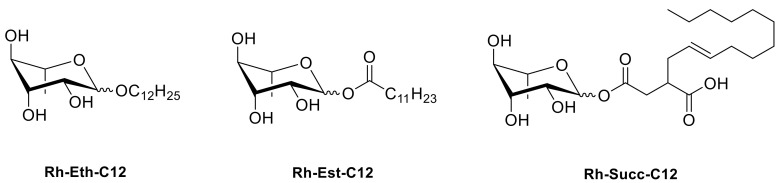
Structure of the three bio-inspired synthetic mono-rhamnolipids (smRLs).

### 4.2. Production and Characterization of the COS BioA187

The chito-oligosaccharide BioA187 was enzymatically extracted and purified from chitosan of fungal cell wall. Fungal chitosan (KitoZyme, Herstal, Belgium) was washed with cold water to remove salts, then dissolved under stirring in 0.1 M acetic acid at room temperature and adjusted to pH 5.5 using NaOH. This was followed by hydrolysis under stirring for 20 h at 30 °C using cellulase from *Trichoderma viride* (VWR, Leuven, Belgium). Purification of the resulting mixture was carried out using tangential flow filtration Vivaflow 200–10,000 MWCO-PES (Sartorius, Goettingen, Germany) followed by a precipitation with 90% ethanol for 48 h, centrifugation and evaporation using rotavap to eliminate remaining ethanol. The pellet was then lyophilized and the BioA187 chito-oligosaccharide was weighted and characterized by Fourier Transform InfraRed (FTIR) and thermogravimetric analysis (TGA). In detail, the thermogravimetric analysis was performed using a TGA 2 (Mettler Toledo) coupled with the STARe Excellence software. A quantity of 3–5 mg of each sample was placed in an open crucible and the temperature was raised from 40 °C to 600 °C at a heating rate of 10 °C per minute. The Fourier Transform InfraRed (FTIR) analyses were performed with a Spectrum One spectrophotometer (Perkin Elmer) operating in transmission mode. FTIR analysis was carried out using the sample dispersed in KBr (1:20 weight ratio). The wavenumber range was 4000–400 cm^−1^ and the resolution was 4 cm^−1^, with 12 scans performed on each sample. The system was coupled with the Spectra software for further analysis, such as obtaining the area of the characteristic peaks (i.e., 1655 cm^−1^ and 1600 cm^−1^ corresponding respectively to amide I and NH2/amide II bands). The deacetylation degree (DD) was determined according to the method reported by Domszy and Roberts [94] using the FTIR results, where A1655 and A3450 are the absorbance at 1655 cm^−1^ of the amide-I band for the measurement of the N-acetyl group content and 3450 cm^−1^ of the hydroxyl band as an internal standard. The factor ‘1.33’ denoted the value of the ratio of A1655/A3450 for fully N-acetylated chitosan (Figure 2).
$$\mathrm{DD}=100-[(\frac{\mathrm{A1655}}{\mathrm{A3450}})\times \frac{100}{1.33}]$$

**Scheme 2 molecules-27-06672-sch002:**
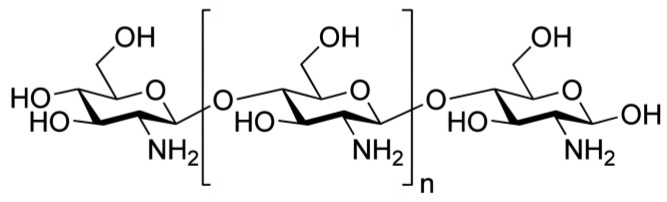
Structure of the chito-oligosaccharide BIOA187.

### 4.3. Production and Purification of LPs

Surfactin, fengycin, mycosubtilin and a mixture of mycosubtilin/surfactin (4/1 *w*/*w*, M + S) powders were produced from *Bacillus subtilis* strains and purified by Lipofabrik (Villeneuve d’Ascq, France) [95]. The fengycin was produced by *B. subtilis* strain Bs2504 [96] and the surfactin was produced by the overproducing *B. subtilis* strain BBG131 [97]. Briefly, surfactin and fengycin were produced in shake flasks (160 rpm) using Landy media according to the growing conditions described by Kourmentza et al. [95]. After centrifugation (8000 g, 30 min), the lipopeptides were purified by two-step ultrafiltration methods on 10 kDa including four steps of diafiltration [95]. After ethanol evaporation, lipopeptides were freeze dried to obtain powder. The mycosubtilin and the mixture M + S were produced by the strain *B. subtilis* LBS1 and the strain BLIP2, respectively, through a semi-industrial scale developed by Lipofabrik [95]. After the culture, the produced lipopeptides were purified using confidential processes and then freeze dried to obtain powders. Finally, a mixture of mycosubtilin/surfactin/fengycin (M + S + F: 33/33/33 *w/w/w*) was realized on demand by Lipofabrik (Villeneuve d’Ascq, France), from the powders of the three lipopeptides previously produced and purified (Figure 3).

**Scheme 3 molecules-27-06672-sch003:**
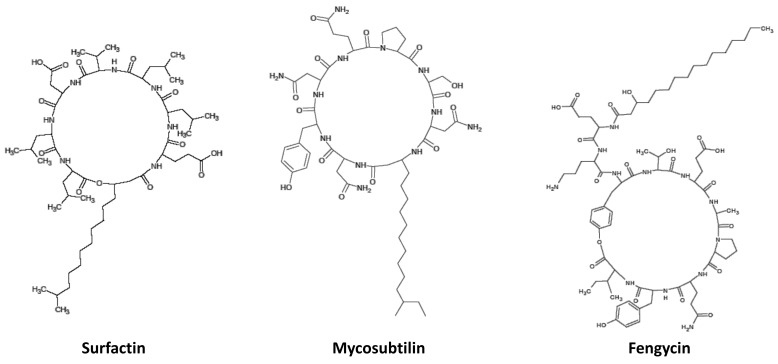
Structure of the cyclic lipopeptides.

### 4.4. Biological Material

Bread wheat (*Triticum aestivum* L.) cv. Pakito, which was provided by RAGT (Rodez, France) was used in these experiments. This cultivar is susceptible to *Bgt*, the causal agent of powdery mildew and exhibits an evaluated field resistance level of 4 on a scale from 1 to 9 (the most susceptible to the most resistant, CTP and Arvalis—Institut du Végétal 2020). Twenty four wheat grains were soaked overnight in water, then grown in trays containing compost (690 g, Jardiland, Calais, France, NPK 18-10-20 à 1.2 kg/m^3^, B: 0.03%, Cu: 0.13%, FeEDTA: 0.09%, Mn: 0.17%, Mo: 0.05% and Zn: 0.04%) and transferred into culture chambers (Panasonic, MLR-352H) with a relative humidity of 70%, a photoperiod of 12 h/12 h, a luminous intensity of 250 μmol × m^−2^ × s^−1^, a day temperature of 18 °C and a night temperature of 12 °C [98]. The *Bgt* MPEbgt1 isolate used in this study was already exploited in precedent works [99,100].

### 4.5. Plant Treatment, Pathogen Inoculation and Disease Evaluation

Three-week-old plants of each tray were treated with 15 mL of each compound or mixture dissolved at 100 µM in dimethyl sulfoxide (DMSO, Honeywell) 0.1% in deionized water, or only with 0.1% of DMSO in deionized water for control plants, by foliar spraying thanks to a sprayer (ITW Surfaces et Finitions, Valence, France) until all leaf area was covered (without going to the flow of the product). Forty-eight hours after spraying, plants were inoculated with 7.5 mL of a solution of *Bgt* spores (500,000 spores·mL^−1^ in Fluorinert FC43, heptacosafluorotributhylamine 3M, Cergy-Pontoise, France) with the same sprayer as before. Twelve days after inoculation, the symptom expression was evaluated by counting the number of white colonies on the third leaf of plantlets. The protection against powdery mildew was determined by assessing the percentage of sporulating colonies on leaves of treated plants compared with control [98].

### 4.6. In Vitro Antifungal Assay

Following a first screening, all molecules were tested at concentrations ranging from 50 to 300 µM. Petri dishes were filled with agar at 12 g·L^−1^, prepared in deionized water, and supplemented with a range of increasing concentrations (50, 100, 200, 300 µM) of each molecule or mixture diluted in 0.1% DMSO. The control being an agar medium containing only 0.1% DMSO. Fresh spores of *Bgt* have been dispersed on the dishes and then observed after 48 h of incubation at 20 °C. The germination was randomly determined on 200 conidia per plate using a light microscope (Olympus BX 40). The conidia were classified in 4 categories: non-germinated spores and spores with aborted tubes (C1), spores with a short appressorial germ tube (AGT, C2), spores with a long AGT (C3), and spores with several AGT (C4) (Figure 6). The spore distribution in the four described categories were reported in frequency of each.

### 4.7. Statistical Analysis

For the protection assays on wheat against *Bgt*, the Shapiro–Wilk test and the Bartlett test were used to verify the normality of the data. Results were analyzed using analysis of variance (ANOVA) followed by multiple comparison by Tukey’s test (*p* < 0.05). To normalize the data presented as percentages, *Bgt* spore germination rates were transformed into angular coordinates (arc sine of the square root: p‘ = arcsin √p) before statistical analysis. Statistical tests were performed with the XLStat software.

## Data Availability

The data presented in this study are available in article.

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
