# Peer review of "Bio-Inspired Rhamnolipids, Cyclic Lipopeptides and a Chito-Oligosaccharide Confer Protection against Wheat Powdery Mildew and Inhibit Conidia Germination"

_molecules, 2022, doi:10.3390/molecules27196672_

Round 1

Reviewer 1 Report

The manuscript entitled “Bio-inspired rhamnolipids, cyclic lipopeptides and a chito-oligosaccharide confer protection against wheat powdery mildew and inhibit conidia germination” presents the evaluation of bioinspired compounds to protect wheat against powdery mildew, caused by the biotrophic Blumeria graminis f.sp. tritici.

The manuscript is very interesting and well-written. However, minor revisions should be made in order to be published in Molecules journal, and the manuscript should be completed and/or modified taking into account the suggestions below:

1.     The authors are advised to rephrase the sentence  from lines 31-34.

2.     The authors are advised to use Italic style for all plant/other species: lines  38, 43, 88, 89, 110,  etc.

3.     The results presented in Fig. 4 could be also listed in a table, for better understanding.

4.     The authors are advise to add details about the synthesis of 3 smRLs with a 12 carbon fatty acid tail (subsection 4.1)

5.     The authors should better explain how they choose the concentrations (line 477)

6.     The authors are advised to check and present references in accordance with suggestions from Instructions for authors

Author Response

Responses to reviewer 1:

We thank reviewer 1 for taking the time to review this article and the various comments made to improve this work. Please find below the different questions/comments of the reviewer and the answers given.

  1. The authors are advised to rephrase the sentence from lines 31-34.

As recommended by the reviewer 1, the sentence (l 31-34) “We also observed a significant inhibitor effect the treatments on the germination of Bgt spore in vitro, with the most anti-germinative activity observed from smRLs Rh-Eth-C12 and Rh-Succ-C12, mycosubtilin and the two CLP mixtures, such molecules could constitute a promising tools for a more eco-friendly agriculture” has been replaced by The higher inhibitor effects  on the germination of Bgt spore in vitro were observed from smRLs Rh-Eth-C12 and Rh-Succ-C12, mycosubtilin and the two CLP mixtures. Taking together, these results revealed that such molecules could constitute promising tools for a more eco-friendly agriculture. ” to clarify the meaning of the phrase, and has been highlighted in yellow.

  1. The authors are advised to use Italic style for all plant/other species: lines 38, 43, 88, 89, 110,  etc.

As recommended by reviewer, the various plants, bacterial or fungal species cited in this article have been checked, written in italics and highlighted in yellow.

  1. The results presented in Fig. 4 could be also listed in a table, for better understanding.

Similar to the questioning of reviewer 1, we wondered about the appropriate way of representing the data on the effect of the products on spore germination. Four different classes (C1 to C4) of spores were studied, these are expressed as frequencies for each condition (percentage), i.e. a cumulative frequency of 100% for each treatment. We chose the representation in the form of stacked histogram in order to be able to compare in a fast way the various tested treatments and the applied concentrations. We think that this representation remains the most interesting to use in our study.

  1. The authors are advise to add details about the synthesis of 3 smRLs with a 12 carbon fatty acid tail (subsection 4.1)

As recommended by reviewer, we add some details concerning the tested smRLs, particularly concerning the step about the quantification and the purity verification of the smRLs synthetized in material and methods part. The steps of smRLs synthetized being clearly explained in the article cited in reference, Robineau et al., 2022.

  1. The authors should better explain how they choose the concentrations (line 477)

The concentrations tested were chosen following preliminaries assays realized on our laboratory. A sentence was added about that in the materiel and methods part.

6.     The authors are advised to check and present references in accordance with suggestions from Instructions for authors

As recommended by reviewer 1, we checked the accuracy of the added references and the way they were presented in accordance with the instructions to authors for this journal.  

Reviewer 2 Report

In this manuscript, The authors studied three different families of natural or bio-inspired compounds derived from bacterial or fungal strains to assess their role in protecting wheat from powdery mildew. Overall, this work is well performed in detailed experimental studies and has implications for the study of bioengineering. I would like to recommend it for publication in Molecules after the following points can be well addressed.

1.     The authors should perform MALDI-MS on the prepared COS BioA187, FTIR and TGA are not sufficient for the chemical properties of the prepared and purified compounds.

2.     Authors should double-check their manuscripts before submitting a revision, there are several expression errors. E.g. “cm-1”, “H2O”, “NH3”,CH3COOH” “250 μmol.m-2.s-1” and line 199-200 “by 45 and 58% compared to the control for M+S or M+F+S, resp”

Author Response

Responses to reviewer 2 :

We thank reviewer 2 for taking the time to review this article and the various comments made to improve this work. Please find below the different questions/comments of the reviewer and the answers given.

  1. The authors should perform MALDI-MS on the prepared COS BioA187, FTIR and TGA are not sufficient for the chemical properties of the prepared and purified compounds.

 As recommended by reviewer 2, the MALDI-MS analysis was performed on COS BioA187, we added a new figure (Figure 3), the corresponding table (Table 1) and a sentence in the results section. In addition, a sentence has also been added in the material and method section. The corrections appear highlighted in yellow

2.     Authors should double-check their manuscripts before submitting a revision, there are several expression errors. E.g. “cm-1”, “H2O”, “NH3”, “CH3COOH” “250 μmol.m-2.s-1” and line 199-200 “by 45 and 58% compared to the control for M+S or M+F+S, resp”

As recommended by the reviewer 2, the manuscript has been carefully proofread to correct various errors, corrections appear highlighted in yellow.